# Molecular structure of human KATP in complex with ATP and ADP

Kenneth Pak Kin Lee[1], Jue Chen[2]*, Roderick MacKinnon[1]*

[1]Laboratory of Molecular Neurobiology and Biophysics, Howard Hughes Medical Institute, The Rockefeller University, New York, United States; [2]Laboratory of Membrane Biology and Biophysics, Howard Hughes Medical Institute, The Rockefeller University, New York, United States

**Abstract** In many excitable cells, KATP channels respond to intracellular adenosine nucleotides: ATP inhibits while ADP activates. We present two structures of the human pancreatic KATP channel, containing the ABC transporter SUR1 and the inward-rectifier $K^+$ channel Kir6.2, in the presence of $Mg^{2+}$ and nucleotides. These structures, referred to as quatrefoil and propeller forms, were determined by single-particle cryo-EM at 3.9 Å and 5.6 Å, respectively. In both forms, ATP occupies the inhibitory site in Kir6.2. The nucleotide-binding domains of SUR1 are dimerized with $Mg^{2+}$-ATP in the degenerate site and $Mg^{2+}$-ADP in the consensus site. A lasso extension forms an interface between SUR1 and Kir6.2 adjacent to the ATP site in the propeller form and is disrupted in the quatrefoil form. These structures support the role of SUR1 as an ADP sensor and highlight the lasso extension as a key regulatory element in ADP's ability to override ATP inhibition.

DOI: https://doi.org/10.7554/eLife.32481.001

## Introduction

In 1983, Akinori Noma made the first recordings of ATP-dependent potassium channels in rat cardiac myocytes (*Noma, 1983*). In membrane patches excised from cardiac myocytes, ATP-dependent $K^+$-channels spontaneously open in electrolyte solutions devoid of nucleotides. When exposed to milimolar concentrations of intracellular ATP, these channels become quiescent. Once ATP is removed, the potassium currents re-emerge, thus ATP facilitates channel closure. KATP channels, as they are known, have since been discovered in other cells, including pancreatic beta cells (*Cook and Hales, 1984*), skeletal (*Spruce et al., 1985*) and smooth muscle (*Standen et al., 1989*) fibers and neurons (*Karschin et al., 1997*; *Nelson et al., 2015*). Through their selectivity for $K^+$ ions KATP channels return membrane voltage toward the resting (Nernst $K^+$) potential (*Hibino et al., 2010*; *Hille, 2001*). In addition to ATP, other co-factors also regulate KATP. $Mg^{2+}$-ADP overrides ATP inhibition (*Dunne and Petersen, 1986*; *Kakei et al., 1986*; *Larsson et al., 1993*; *Nichols et al., 1996*) and phosphatidyl-inositol diphosphate ($PIP_2$) is required for activity (*Baukrowitz et al., 1998*; *Hilgemann and Ball, 1996*; *Shyng and Nichols, 1998*). By virtue of their sensitivity to both ATP and ADP, these channels are thought to couple cellular metabolism to membrane excitability.

The physiological importance of KATP channels is underscored by mutations associated with heritable human diseases including congenital hyperinsulinism (*Pinney et al., 2008*; *Saint-Martin et al., 2011*; *Sharma et al., 2000*) and permanent neonatal diabetes mellitus (*Aittoniemi et al., 2009*; *Ashcroft et al., 2017*; *Letha et al., 2007*). Furthermore, drugs used in the treatment of diabetes mellitis, hypertension and alopecia target KATP channels (*Aziz et al., 2014*; *Feldman, 1985*; *Rubaiy, 2016*; *Shorter et al., 2008*; *Standen et al., 1989*; *Sturgess et al., 1985*).

A broad and rich body of literature based on decades of study describes the biochemical and functional properties of KATP channels (*Aguilar-Bryan and Bryan, 1999*; *Aguilar-Bryan et al., 1998*; *Ashcroft et al., 2017*; *Ashcroft and Ashcroft, 1992*; *Hibino et al., 2010*; *Nichols, 2016*;

*For correspondence:
juechen@rockefeller.edu (JC);
mackinn@rockefeller.edu (RMK)

Competing interests: The authors declare that no competing interests exist.

**eLife digest** A hormone called insulin finely controls the amount of sugar in the blood. When the blood sugar content is high, a group of cells in the pancreas release insulin; when it is low, they stop. In these cells, the level of sugar in the blood modifies the ratio of two molecules: ATP, the body's energy currency, and ADP, a molecule closely related to ATP. Changes in the ATP/ADP ratio are therefore a proxy of the variations in blood sugar levels.

In these pancreatic cells, a membrane protein called ATP sensitive potassium channel, $K_{ATP}$ channel for short, acts as a switch that turns on and off the production of insulin. ATP and ADP control that switch, with the two molecules having opposite effects on the channel – ATP deactivates it, ADP activates it. The changes in ATP/ADP ratio – and by extension in blood sugar levels – are therefore coupled with the release of insulin.

However, how $K_{ATP}$ channels sense the changes in the ATP/ADP ratio in these cells is still unclear. In particular, ATP levels are usually high and constant: ATP is then continuously deactivating the channels, and it is unclear how ADP ever activates them.

Here, Lee et al. use a microscopy technique that can image biological molecules at the atomic scale to look at the structure of human pancreatic $K_{ATP}$ channels. The 3D reconstruction maps show that $K_{ATP}$ channels have binding sites for ATP but also one for ADP. This ADP site acts as a sensor that can detect even small changes in ADP levels in the cell. The maps also reveal a dynamic lasso-like structure connecting the ATP and ADP binding areas. This domain may play a vital role in allowing ADP to override ATP's control of the channel. The presence of the ADP sensor and the lasso structure could explain how $K_{ATP}$ channels monitor changes in the ATP/ADP ratio and can therefore control the release of insulin based on blood sugar levels.

Defects in the $K_{ATP}$ channels of the pancreas are present in genetic diseases where infants produce too much or too little insulin. Understanding the structure of these channels and how they work may help scientists to design new drugs to treat these conditions.

DOI: https://doi.org/10.7554/eLife.32481.002

*2006*). These channels were found to be large oligomeric complexes composed of four sulphony-lurea receptor (SUR) subunits, which belong to the ABCC family of ABC transporters, and four Kir6 subunits, which are members of the inward rectifier potassium channel family (*Clement et al., 1997*; *Inagaki et al., 1997*; *Shyng and Nichols, 1997*). This peculiar combination of an ABC transporter and an ion channel has been the focus of extensive study and speculation.

Recent cryo-EM structures of hamster SUR1-mouse/rat Kir6.2 hybrid complexes presented the architecture of KATP, revealing a central $K^+$ channel surrounded by four SURs (*Li et al., 2017*; *Martin et al., 2017a*, *2017b*). The SURs extended away from the channel, akin to propellers, and their nucleotide-binding domains (NBDs) were empty and dissociated from each other. In this study, we analyze human KATP with adenosine nucleotides bound to SUR1.

## Results and discussion

### Two structures of human KATP

In humans, there are two SUR genes (SUR1 and SUR2) and two Kir6 genes (Kir6.2 and Kir6.1) (*Aguilar-Bryan et al., 1995*; *Babenko et al., 1998*; *Chutkow et al., 1996*; *Inagaki et al., 1995a*, *Inagaki et al., 1995b*, *Inagaki et al., 1995c*). The focus of this study is the human KATP channel composed of SUR1 and Kir6.2, which are found in pancreatic beta cells where they play a major role in regulating insulin secretion. To facilitate large-scale expression and purification of the human SUR1-Kir6.2 complex, we fused the C-terminus of SUR1 to the N-terminus of Kir6.2 using a six-amino acid linker containing three repeats of the Ser-Ala dyad. This strategy allows the production of human SUR1-human Kir6.2 complex as a tetramer of the fusion construct. This and other peptide linkers ranging from 6 to 14 residues have been used to generate KATP channel fusions that reproduce the electrophysiological and pharmacological characteristics of wild-type octameric KATP channels, as assessed by inhibition from ATP and sulphonylurea drugs as well as activation by ADP and diazoxide (*Chan et al., 2008*; *Clement et al., 1997*; *Mikhailov et al., 2005*, *1998*). We also

confirmed functional expression of our fusion construct (KATP$_{em}$) in electrophysiology recordings and ATPase measurements (*Figure 1—figure supplement 1*). For structural studies, KATP$_{em}$ was purified in PMAL-C8 and mixed with Mg$^{2+}$, ATP, vanadate, and C8-PIP$_2$ before imaging.

Cryo-EM reconstructions revealed two major conformations (*Figure 1A*). One is similar to the published structure ('propeller' form) with the exception that the NBDs are closed in our structure (*Figure 1C*). The other conformation, which we refer to as the quatrefoil form, is dramatically different (*Figure 1B*). The NBDs are also closed in the quatrefoil form; however, the SUR1 subunits reside in a different location relative to Kir6 (*Figure 1B and C*). Upon further classification and refinement, the propeller form was determined to 5.6 Å resolution and the quatrefoil form was determined to 3.9 Å resolution.

The EM map of the quatrefoil form was further improved by focused classification and symmetry expansion as implemented in RELION (see Materials and methods)(*Scheres, 2016*). Using this approach, we obtained local maps with substantially enhanced resolution, from which we pieced together a composite 3D image of the KATP$_{em}$ particle (*Figure 1B*). This enabled building of 1596 out of 1977 residues in the complex and assignment of the amino acid register (*Figure 2*). In addition, the map was of sufficient quality to identify ligands including Mg$^{2+}$, ATP, and ADP.

In the quatrefoil form, KATP is a symmetrical tetramer; each protomer consists of one K$^+$ channel subunit and one ABC transporter. There are five domains in the ABC transporter, TMD0, TMD1, NBD1, TMD2 and NBD2 (*Figure 2A*). The latter four domains form the transporter module. The K$^+$ channel consists of a transmembrane domain and a cytoplasmic domain (CTD), which form the ion pathway through the complex (*Figure 2B*). At the molecular center, the four subunits of Kir6.2 form a canonical inward-rectifier K$^+$ channel structure (*Figures 1B* and *2B*). The TMD0s are bound to the channel and hold the four ABC transporters akin to the leaves of a quatrefoil. EM density of the entire complex is very well defined, but residues 193–261, known as the L0-loop, containing a conserved lasso motif (*Johnson and Chen, 2017*; *Li et al., 2017*; *Martin et al., 2017b*; *Zhang and Chen, 2016*) is not visible (*Figure 2*).

A focused 3D classification and refinement strategy coupled with symmetry expansion was also employed to improve the propeller form reconstruction (*Figure 1—figure supplement 6*). Although the final propeller form EM map is of lower resolution than that of the quatrefoil form, secondary structural elements are well defined (*Figure 1C*). Each domain of the KATP maintains the same structure in both quatrefoil and propeller forms (*Figure 2—figure supplement 1*). The major difference between the two structures lies in the different positions of the transporter module relative to the molecular center (*Figure 1B and C*; *Figure 2—figure supplement 1*). In addition, the L0-loop is clearly visible in the propeller form albeit at lower resolution (*Figure 1C* and *Figure 1—figure supplement 7*).

We further note that the distance between the C-terminus of SUR1 and the first visible residue of Kir6.2 (Arg32) in the propeller form is 64.2 Å, which is even greater than the corresponding distance in a recent propeller KATP structure without bound Mg$^{2+}$-nucleotides (59.4 Å, PDB ID: 5TWV) (*Martin et al., 2017b*). Assuming the Cα-Cα distance of adjacent residues in an extended strand spans 3.5 Å (*Berg et al., 2002*), the maximum distance spanned by the disordered region connecting the C-terminus of SUR1 and Kir6.2 Arg32 (37 a.a. total, including the (SA)$_3$ linker) is 118.4 Å, which exceeds the observed distances mentioned above by ~2-fold. Thus, we do not expect the six-amino acid linker used in the KATP$_{em}$ construct to limit the range of motion sampled by SUR1 and Kir6.2 or introduce unnatural distortions to the propeller or quatrefoil forms. Moreover, the fusion construct is functional and sensitive to nucleotides (*Figure 1—figure supplement 1*).

## The nucleotide-bound state of SUR1

The transporter module of SUR1 resembles a canonical ABC transporter, TMD1 and TMD2 are domain-swapped, with TM9-10 from TMD1 and TM15-16 from TMD2 reaching across the interface between half-transporters to pack against the neighboring TMD (*Figure 3A*). The NBDs dimerize in a head-to-tail fashion and nucleotides occupy both ATPase active sites at the dimer interface.

In an ABC transporter that functions through the principle of alternating access (*Jardetzky, 1966*), a translocation pathway opens to the extracellular space upon NBD dimerization (*Dawson and Locher, 2007*). By contrast, in the NBD-dimerized SUR1, a small cavity inside the TMDs is closed off from the extracellular milieu and remains accessible from the cytoplasm (*Figure 3A*). These features are consistent with SUR1 being a regulator of a K$^+$ channel rather than a *bona fide* transporter itself.

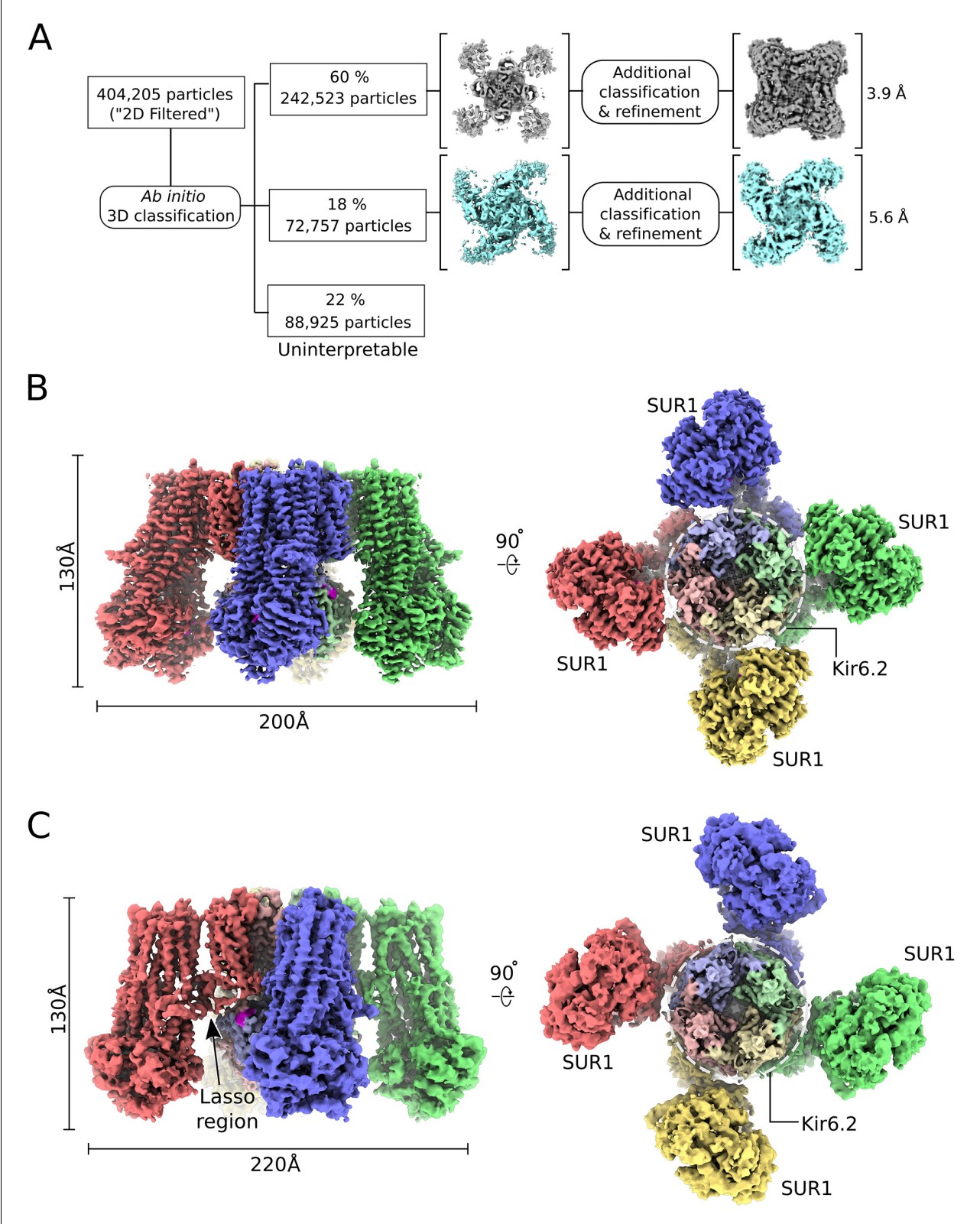

**Figure 1.** CryoEM reconstruction of the human KATP channel. (**A**) Ab-initio 3D classification separates KATP$_{em}$ particles into two distinct structures - the quatrefoil form (grey) and the propeller form (light blue). (**B**) EM density map of the KATP channel quatrefoil form. Left – Sideview of the EM density map. Right – cytoplasmic view of the EM density map. The CTDs of KIR6.2 are visible in this orientation. SUR1-KIR6.2 fusion protomers are colored red,

*Figure 1 continued on next page*

*Figure 1 continued*

green, blue or yellow, respectively. The final symmetrized composite map is shown. (C) EM density map of the KATP channel propeller form. See also Figure 1-figure supplements 1 to 7.

DOI: https://doi.org/10.7554/eLife.32481.003

The following figure supplements are available for figure 1:

**Figure supplement 1.** Functional characterization of the human KATP channel.

DOI: https://doi.org/10.7554/eLife.32481.004

**Figure supplement 2.** Flow chart of the image processing workflow.

DOI: https://doi.org/10.7554/eLife.32481.005

**Figure supplement 3.** CryoEM reconstruction of the quatrefoil form KATP channel.

DOI: https://doi.org/10.7554/eLife.32481.006

**Figure supplement 4.** Focus maps of the quatrefoil form KATP channel used for composite map generation.

DOI: https://doi.org/10.7554/eLife.32481.007

**Figure supplement 5.** Cryo-EM densities of different regions of the quatrefoil form KATP channel.

DOI: https://doi.org/10.7554/eLife.32481.008

**Figure supplement 6.** Focus maps of the propeller form KATP channel used for composite map generation and comparison with original map.

DOI: https://doi.org/10.7554/eLife.32481.009

**Figure supplement 7.** Cryo-EM densities of different regions of the propeller form KATP channel.

DOI: https://doi.org/10.7554/eLife.32481.010

This notion is further supported by the observation that no substrate has been identified for SUR1 hitherto (*Aittoniemi et al., 2009*; *Schwappach et al., 2000*). Whether the transporter transitions to the outside-open conformation as part of the KATP gating cycle awaits further investigation.

Another unusual feature of SUR1 is the asymmetric configuration of the NBD dimer. $Mg^{2+}$-ATP is bound in the 'closed' catalytically inactive degenerate site and $Mg^{2+}$-ADP is bound in the 'open' catalytically competent consensus ATPase site (*Figure 3B–D*, *Figure 3—figure supplement 1*). To our knowledge, this is the only ABC transporter observed with both $Mg^{2+}$-ADP and $Mg^{2+}$-ATP bound in its NBDs. We note that the structure of a dimeric ABC transporter, AaPrtD, shows $Mg^{2+}$-ADP bound to both sites, but in this case both sites are the same and symmetrically closed around $Mg^{2+}$-ADP (*Figure 3—figure supplement 2*). In the structure determination of KATP, no ADP was added to the sample. While some ADP must have been generated through hydrolysis, it is clear that the degenerate ATP-binding site on SUR1 and the ATP inhibitory site on the channel are occupied by ATP. Thus, the consensus site on SUR1, relative to the other ATP sites in the complex, is selective for ADP (see below).

A simplified reaction scheme for the consensus ATP site (S) in SUR1 can be written as S + ATP $\longleftrightarrow$ SATP $\longleftrightarrow$ S'ATP$\rightarrow$ S'ADP +Pi $\longleftrightarrow$ S'+ADP. Here, the hydrolysis reaction is approximated as irreversible. We know from the rate of ATP hydrolysis (*Figure 1—figure supplement 1C*) that the ATP turnover rate, expressed per SUR1 molecule, is approximately 0.02 per second, which means the above reaction takes approximately 50 s to undergo a complete cycle. The structure shows a relatively open cleft between bound ADP and solution, therefore ADP dissociation is likely to be much faster than this slow turnover rate of 0.02 per second. A different transition must limit the overall rate. The slow transition is likely to be either the isomerization step from NBD open to NBD closed conformation (SATP $\longleftrightarrow$ S'ATP) or the hydrolysis step (S'ATP$\rightarrow$ S'ADP + Pi). Base on this reasoning, the occupancy of S' by ADP must reflect equilibration of the site by ADP in solution (i.e. through a rapid, reversible ADP dissociation reaction, S'ADP $\longleftrightarrow$ S' + ADP). The prior observation that channel gating does not exhibit violation of microscopic reversibility is also compatible with this assertion (*Choi et al., 2008*). We will return to the significance of this observation and its consistency with prior studies (*Dunne and Petersen, 1986*; *Dunne et al., 1988*; *Kakei et al., 1986*; *Nichols et al., 1996*; *Tantama et al., 2013*) when we consider the mechanism KATP's ability to sense the metabolic state of a cell.

SUR1 is the target of sulphonylureas that inhibit KATP to promote insulin secretion (*Aguilar-Bryan et al., 1995*; *Ashcroft et al., 1987*; *Bryan et al., 2005*; *Dean and Matthews, 1968*; *Sturgess et al., 1985*). In a recent structure of the hamster-rat KATP hybrid, the glibenclamide-binding site was shown to reside within the TMDs of the inward-facing (i.e. NBD open) transporter module (*Martin et al., 2017a*)(*Figure 3E*). In the NBD-dimerized form, residues that comprise the

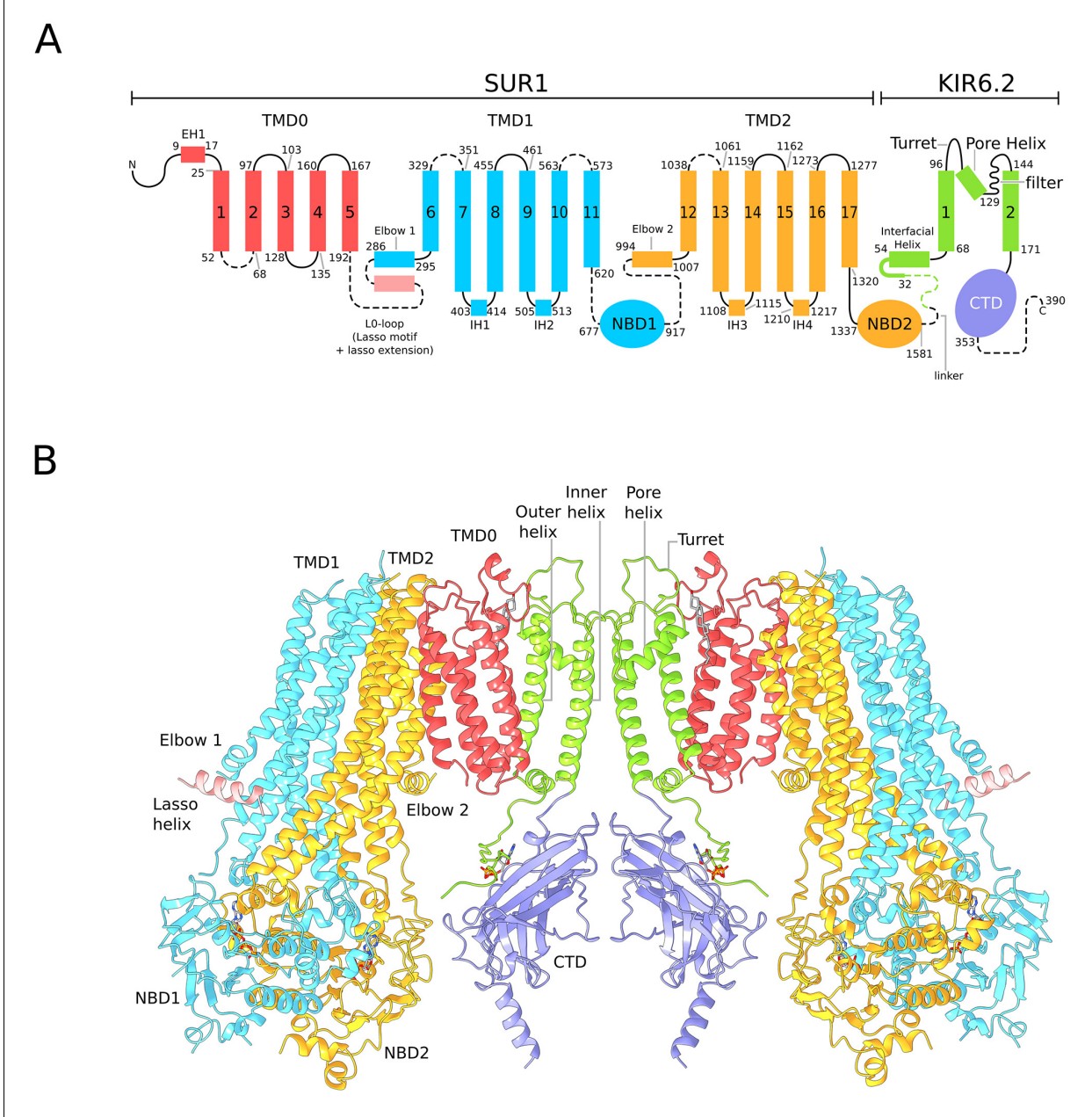

**Figure 2.** Architecture of the KATP channel. (**A**) Domain architecture of the SUR1-KIR6.2 fusion construct with a flexible six-amino acid linker joining the C-terminal end of SUR1 to the KIR6.2 N-terminus. The numbering scheme conforms to the native human SUR1 and human KIR6.2 sequences. Dashed lines denote regions where clear density was not observed in the quatrefoil form reconstruction. The lasso motif and lasso extension is visible in the propeller form but not the quatrefoil. (**B**) Ribbons representation of two SUR1-KIR6.2 fusion protomers in the quatrefoil form.

DOI: https://doi.org/10.7554/eLife.32481.011

The following figure supplement is available for figure 2:

**Figure supplement 1.** Comparison of quatrefoil and propeller forms.

DOI: https://doi.org/10.7554/eLife.32481.012

glibenclamide-binding pocket move closer towards each other, making the cavity too small to accommodate glibenclamide (*Figure 3E*). This observation is entirely consistent with the hypothesis that glibenclamide stabilizes the inward-facing conformation (*Martin et al., 2017a*). It functions as a wedge to prevent adenosine nucleotide-mediated NBD closure and thus prevents SUR-mediated regulation of channel activity.

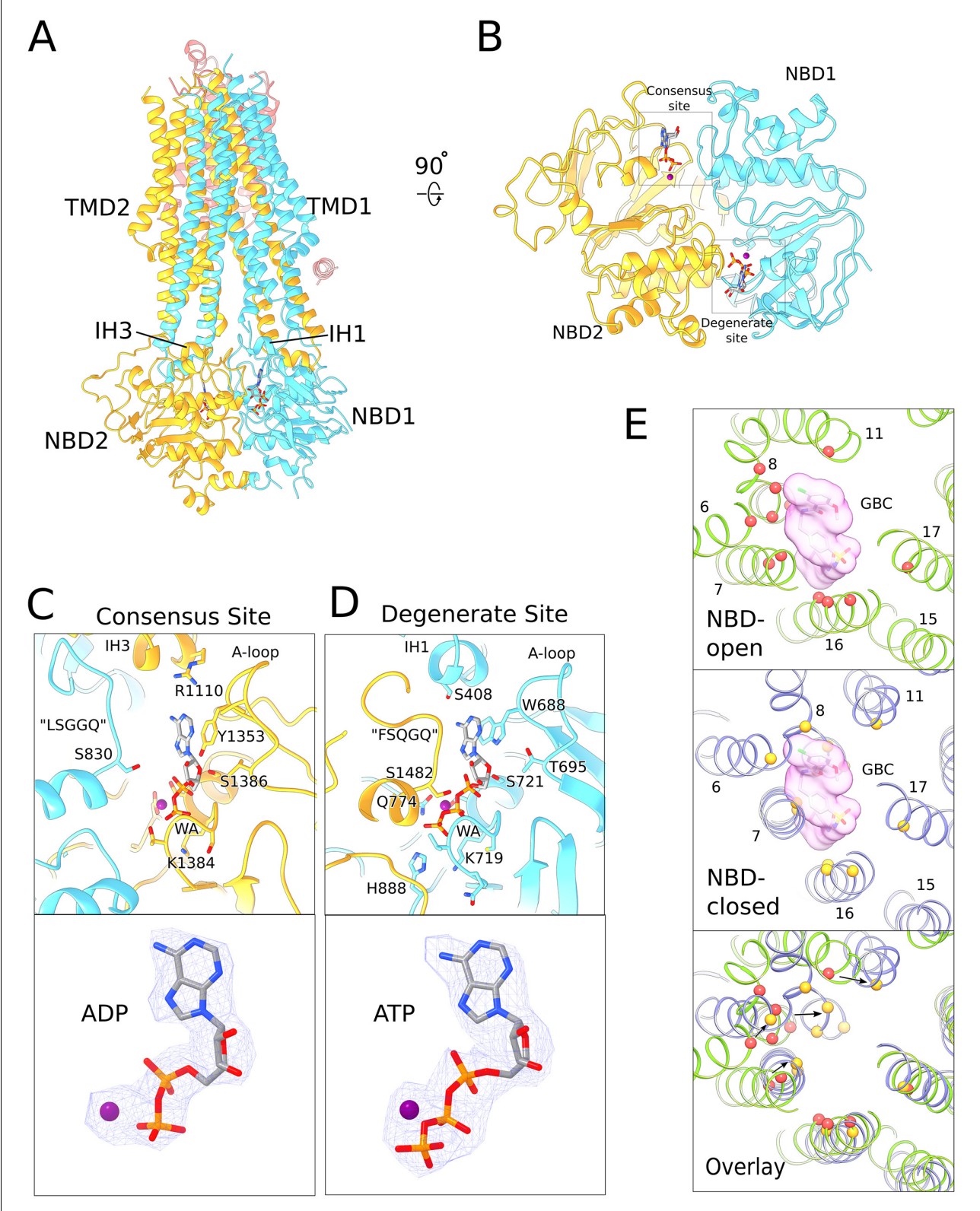

**Figure 3.** Structure of human SUR1 in complex with Mg$^{2+}$-ATP and Mg$^{2+}$-ADP. (**A**) Ribbons representation of the SUR1 subunit. Color scheme: TMD0, pink; TMD1-NBD1, blue; TMD2-NBD2, yellow. Mg$^{2+}$-ADP and Mg$^{2+}$-ATP are shown in stick model. (**B**) The NBD dimer viewed from the membrane. Mg$^{2+}$-ADP and Mg$^{2+}$-ATP bind to the consensus and degenerate ATPase sites, respectively, to generate an asymmetric NBD dimer. (**C**) Close-up view of the SUR1 consensus ATPase site. Top - Residues in contact with Mg$^{2+}$-ADP are shown. The signature 'LSGGQ' motif in NBD1 is disengaged from the

*Figure 3 continued on next page*

Figure 3 continued

bound nucleotide, resulting in a relatively 'open' consensus ATPase site. Bottom - shows EM density corresponding to $Mg^{2+}$-ADP as a blue mesh. (D). Close-up view of the SUR1 degenerate ATPase site. Top - Residues in contact with $Mg^{2+}$-ATP are shown. The signature sequence in NBD2 is mutated to 'FSQGQ' and makes direct contacts with the bound ATP molecule. This results in a 'closed' degenerate ATPase site. Bottom - shows EM density corresponding to $Mg^{2+}$-ATP as a blue mesh. (E) Structural changes in the glibenclamide (GBC)-binding pocket in the NBD-open and NBD-closed states. Top panel – The GBC-binding pocket mapped to the NBD-open SUR1 structure (PDB: 6BAA). Residues in contact the GBC are indicated by red spheres (*Martin et al., 2017b*). A surface representation of the modeled GBC molecule is shown (pink). Middle panel – GBC binding residues (yellow spheres) mapped onto the NBD-closed SUR1 in the quatrefoil form. GBC is shown to indicate potential steric clashes between the inhibitor and the transporter. Bottom panel – Superposition of the NBD-open and NBD-closed states of SUR1 showing the displacement of GBC-binding residues upon nucleotide binding. The superposition shown was obtained by alignment of the half-transporter from the NBD-closed and NBD-open SUR1 comprised of: TM9,10,12–14,17 and NBD2. Unless stated otherwise, the quatrefoil form is displayed in all panels.

DOI: https://doi.org/10.7554/eLife.32481.013

The following figure supplements are available for figure 3:

**Figure supplement 1.** Comparison of ATP and ADP densities.
DOI: https://doi.org/10.7554/eLife.32481.014

**Figure supplement 2.** Comparison of NBD-closed SUR1 and ADP bound AaPrtD.
DOI: https://doi.org/10.7554/eLife.32481.015

## The inhibitory ATP-binding site of human Kir6.2

EM density for the Kir6.2 channel is shown in *Figure 4A* to illustrate the quality of the map in this region of the transporter-channel complex. Excellent side chain density is observed in both the pore and CTD, which have allowed us to build most of the channel except for disordered flexible regions in the N- and C-termini of Kir6.2 (*Figure 4B*).

The overall structure of human Kir6.2 is that of an inward-rectifier $K^+$-channel with a large cytoplasmic domain. In this structure, both the inner helix gate and the G-loop gates are closed (*Figure 4B* and *Figure 4—figure supplement 1*). No $PIP_2$ density was observed in the putative $PIP_2$ binding sites even though C8-$PIP_2$ was present in the sample. By contrast $PIP_2$ was present in Kir2 and GIRK inward rectifiers at similar concentrations (*Hansen et al., 2011*; *Whorton and MacKinnon, 2011*). Its absence in the KATP structure suggests that ATP, which is known to negatively regulate $PIP_2$-mediated activation, may have prevented $PIP_2$ binding (*Baukrowitz et al., 1998*; *Shyng and Nichols, 1998*).

Four ATP molecules are associated with the Kir6.2 channel (one ATP per subunit); each binds in a shallow pocket on the surface of the CTD (*Figure 4*). Density for ATP is unambiguous (*Figure 4D*). Residues N48 and R50 from a neighboring subunit make two hydrogen bonds with the Watson-Crick edge of the adenine base. These interactions likely account for the specific recognition of ATP versus GTP (*Schwanstecher et al., 1994*; *Tucker et al., 1998*)(*Figure 4C*). This same principle of nucleotide selectivity is observed in protein kinases in which main chain atoms of an inter-domain linker decode the Watson-Crick edge of the incoming ATP co-substrate (*Knighton et al., 1991*). Out of the 14 residues that constitute the ATP binding site, mutation of seven are associated with diabetes mellitus, underscoring the physiological importance of ATP regulation (*Edghill et al., 2010*; *Gloyn et al., 2004*; *Lang and Light, 2010*; *Proks et al., 2004*). Notably, G334 and N335 are uniquely found in the Kir6 members of the inward rectifier family (*Figure 4—figure supplement 2*). In GIRK, the residue analogous to G334 is replaced by histidine, which likely prevents ATP binding by steric hindrance (*Drain et al., 1998*; *Masia et al., 2007*). These features help to explain why among inward rectifiers Kir6 is uniquely sensitive to ATP.

## ATP-binding site is a nexus for communication between regulatory ligands

Three naturally occurring ligands control the activity of KATP channels: $PIP_2$ is required for activity (*Hilgemann and Ball, 1996*), ATP inhibits activity (*Inagaki et al., 1995a*; *Noma, 1983*; *Rorsman and Trube, 1985*), and ADP potentiates activity (*Dunne and Petersen, 1986*). All three of these ligands act in a dependent fashion. ATP in functional experiments appears to negatively regulate the action of $PIP_2$ (*Baukrowitz et al., 1998*; *Shyng and Nichols, 1998*) and ADP appears to override ATP inhibition (*Dunne and Petersen, 1986*; *Kakei et al., 1986*; *Larsson et al., 1993*; *Nichols et al., 1996*).

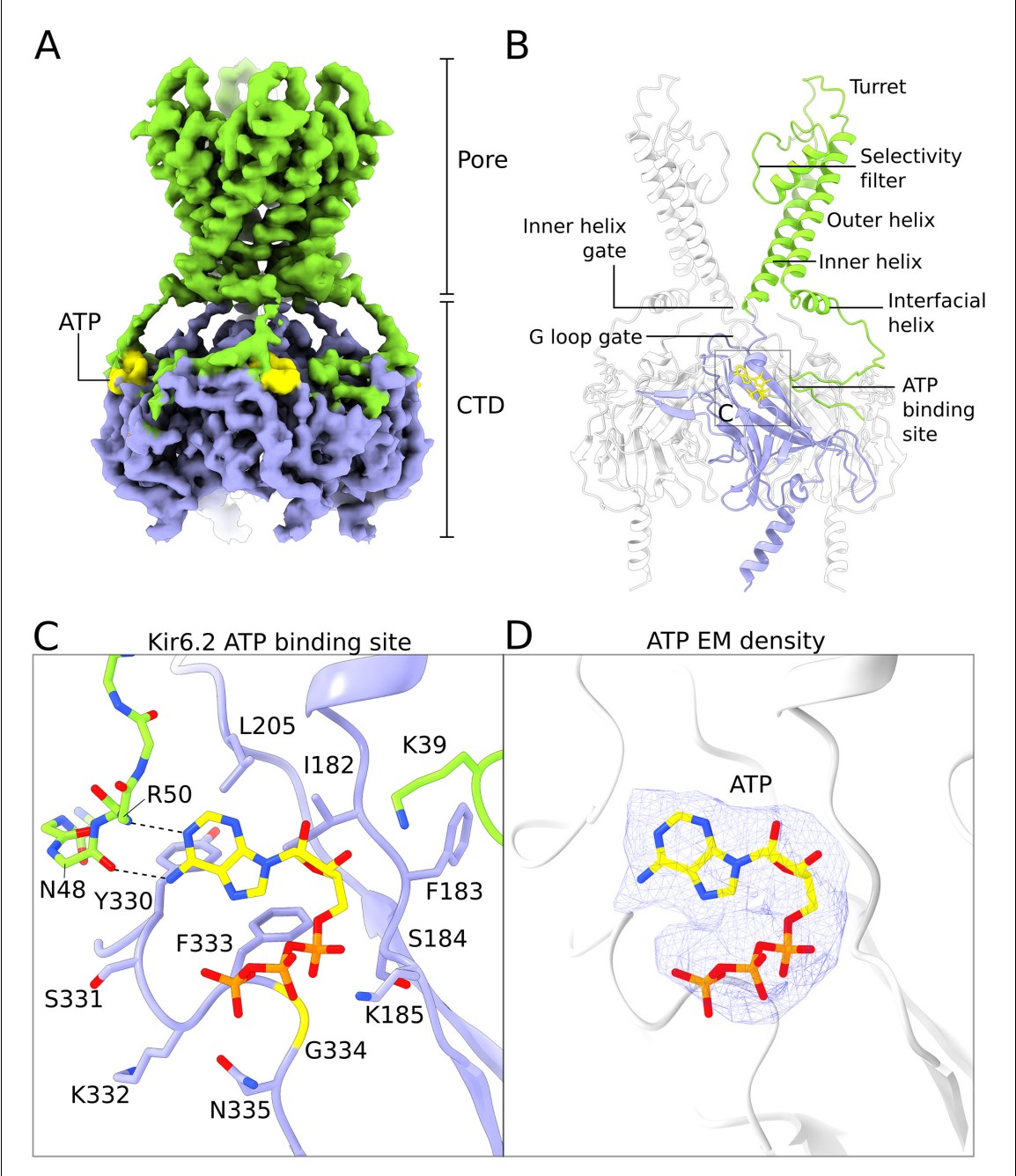

**Figure 4.** Structure of human Kir6.2 bound to ATP. (**A**) EM density of the Kir6.2 tetramer. Sidechain density is clearly observed and β-strands are well resolved. EM densities corresponding to bound ATP molecules are colored yellow. (**B**) Ribbons representation of the Kir6.2 atomic model. For clarity only two pore domains are shown. The CTD domain in the back is also omitted. One Kir6.2 subunit is colored according to the coloring scheme using in *Figure 2A*. Important structural elements are indicated. The ATP molecule bound to the colored Kir6.2 subunit is shown as a yellow ball-and-stick model. (**C**) Close-up view of the Kir6.2 ATP binding site. Residues in contact with the bound ATP molecule are shown. The N-terminal extension of the interfacial helix from a neighboring Kir6.2 subunit contacts the purine base of ATP via mainchain interactions and is shown in ball-and-stick representation. Hydrogen-bonding interactions between mainchain atoms in the Kir6.2 N-terminus and the adenine base are shown as dashed lines. ATP is shown as a ball-and-stick model, with carbon atoms colored yellow. (**D**) EM density corresponding to the bound ATP molecule, shown as a blue mesh. ATP is displayed in the same fashion as in (**C**). The unusual horseshoe-shaped conformation of ATP is evident. The quatrefoil form is displayed in all panels.

DOI: https://doi.org/10.7554/eLife.32481.016

The following figure supplements are available for figure 4:

*Figure 4 continued*

**Figure supplement 1.** Comparison of Kir6.2 and GIRK ion conduction pathways.
DOI: https://doi.org/10.7554/eLife.32481.017

**Figure supplement 2.** Sequence alignment of KIR channels.
DOI: https://doi.org/10.7554/eLife.32481.018

We observe a structural interconnectedness of the ATP-binding site, $PIP_2$-binding site and SUR - which is the seat of ADP binding - that seems relevant to their functional dependence.

One side of the inhibitory ATP-binding site on Kir6.2 is formed by the N-terminal polypeptide segment that leads to the interfacial 'slide' helix (*Figures 4C* and *5A*). A local structural comparison of Kir6.2 and GIRK (*Whorton and MacKinnon, 2011*) shows that these structural elements are shifted in Kir6.2, presumably to accommodate binding of ATP as residues N48 and R50 reside in this region (*Figures 4C* and *5A*, *Figure 5—figure supplement 1*). The shift compresses the $PIP_2$-binding site in Kir6.2 compared to GIRK. This compression, if it occurs dynamically when ATP binds, offers a plausible explanation for ATP inhibition through competition with the essential ligand $PIP_2$.

While ATP inhibition is intrinsic to Kir6.2 (*Tucker et al., 1997*), SUR1 exerts two opposing influences on ATP inhibition. First, the presence of SUR1 (compared to Kir6.2 expressed in its absence) enhances the potency of ATP inhibition (*Tucker et al., 1997*). Second, SUR1 permits $Mg^{2+}$-ADP to override ATP inhibition (*Nichols et al., 1996*; *Tucker et al., 1997*). The structures of the quatrefoil and propeller forms provide clues as to how SUR1 might accomplish these tasks.

In the propeller form the ATP-binding site on Kir6.2 is buttressed by the lasso extension, which attaches SUR1 to the CTD (*Figure 5B*). Because of the lasso extension's close proximity to the ATP-binding site it might influence ATP binding. This could account for the fact that SUR1 coexpression potentiates ATP inhibition of KATP by a factor more than 10-fold (*Tucker et al., 1997*). It could also provide the structural pathway through which ADP binding to SUR overcomes ATP inhibition to activate KATP. Functional data are consistent with this possibility. The lasso extension and adjacent amino acids within the ATP-binding site (i.e. both sides of the interface between Kir6.2 and SUR1) are hotspots for inherited gain-of-function mutations, which cause diabetes owing to over-activity of KATP channels (*Edghill et al., 2010*; *Gloyn et al., 2004*; *Lang and Light, 2010*; *Proks et al., 2004*). Furthermore, when this interface is locked together by a disulfide cross-link, KATP is permanently inhibited (*Pratt et al., 2012*) (*Figure 5B*). While a mechanistic description of how SUR regulates Kir6 is still unknown, the structure of the lasso extension/ATP-binding site interface and its correlation to disease-causing mutations underscores its potential importance.

## Implications for channel regulation

Formation of the quatrefoil form is associated with disruption of the lasso extension/ATP-binding site interface. SUR1 remains attached to Kir6.2 through its TMD0-channel interface, but rotation of the transport module around the central axis of TMD0 (to convert from propeller to quatrefoil forms) can only occur if the lasso extension releases from its contact adjacent to the ATP site on the CTD. We do not know whether disruption of this interface is an artifact of the preparation or is a normal occurrence during channel gating. Our rationale for even considering the latter possibility is based on a striking difference between the propeller and quatrefoil structures depicted in *Figure 6*. In the propeller form, in which the lasso extension/Kir6 interface is intact, the SUR1 transporter modules are held with an offset with respect to the hydrophobic membrane plane, as defined by the position of the Kir6.2 channel and TMD0 domains (*Figure 6—figure supplement 1*). In the quatrefoil form, the SUR1 transport modules reside coplanar, that is, in the same hydrophobic plane as defined by the Kir6 and TMD0 elements (*Figure 6*). The consequence of holding the transport modules at a shifted position with respect to the pore in the propeller form should be strain exerted on the lasso extension/Kir6 interface in the propeller form.

When we compare the NBD-dimerized propeller form in this study with the published NBD-open propeller form (*Li et al., 2017*; *Martin et al., 2017b*), we observe that dimerization causes a slight tilting of NBD1 away from the center of the complex (*Figure 6—figure supplement 2*). Thus, dimerization appears to exert force on the lasso extension away from the channel. In this manner, nucleotide binding and dimerization of the NBDs could potentially interfere with ATP inhibition. Whether the lasso extension actually dissociates is unknown. The gain-of-function mutations on the lasso

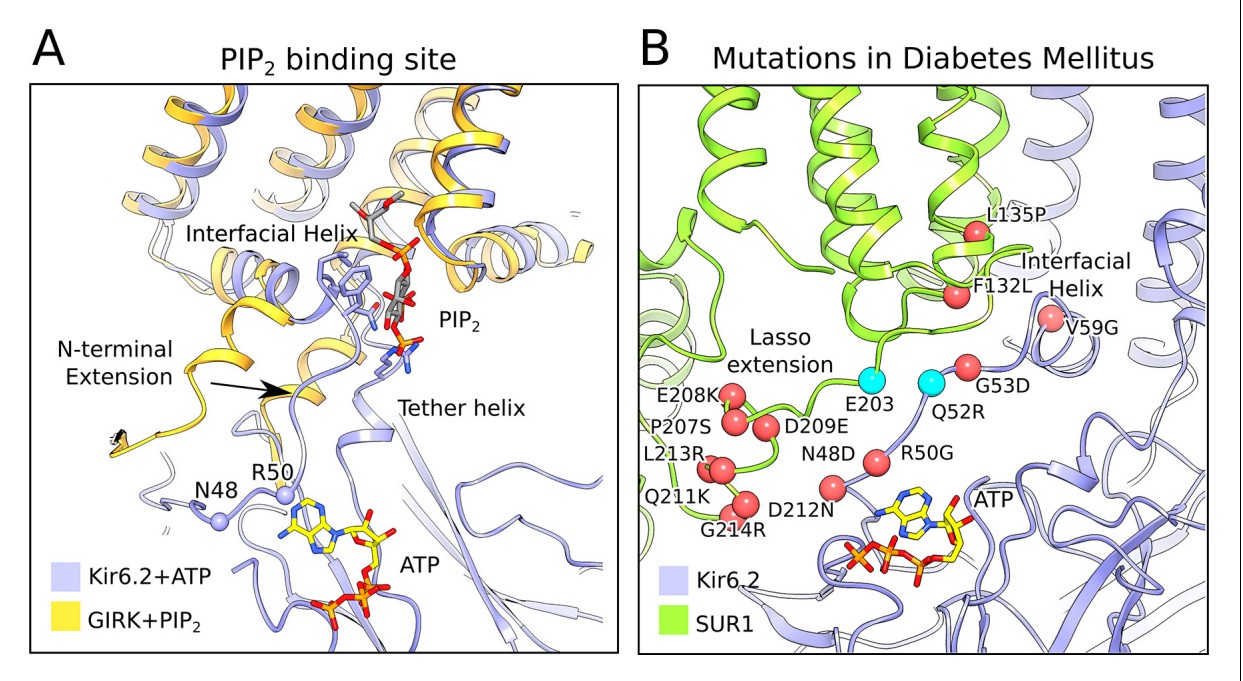

**Figure 5.** The PIP$_2$-binding site and mutations associated with neonatal diabetes. (**A**) Superposition of the ATP-bound Kir6.2 with PIP$_2$-bound GIRK (PDB: 3SYA), showing the linkage between the two ligand binding sites. The arrow indicates the movement of the N-terminal extension from the interfacial helix, as compared to GIRK, that accompanies ATP binding. Structural overlay was obtained by alignment of the selectivity-filter sequences of GIRK and Kir6.2. The quatrefoil form of KATP$_{em}$ is shown. (**B**) A close-up view of the Lasso extension-Kir6.2 interface. SUR1 and Kir6.2 in the propeller form are shown in ribbons representation and are colored in green and blue, respectively. Mutations associated with diabetes are indicated by red spheres. Cyan spheres indicate the location of residues E203 in SUR1 and Q52 in Kir6.2, which irreversibly inhibits channel opening when cross-linked via a disulfide-bridge (**Pratt et al., 2012**). The model shown corresponds to the propeller form of KATP$_{em}$.

DOI: https://doi.org/10.7554/eLife.32481.019

The following figure supplement is available for figure 5:

**Figure supplement 1.** Comparison of PIP$_2$-binding site in KIR channels.

DOI: https://doi.org/10.7554/eLife.32481.020

extension that should tend to disrupt the interface, and the cross-link that should stabilize it, seem to support the idea that the interface is dynamic (**Edghill et al., 2010**; **Lang and Light, 2010**; **Pratt et al., 2012**). Whether or not it is dynamic to the extent that it relinquishes its contact with the channel, we think it likely that dimerization of the NBDs acts through the lasso extension to activate the channel.

The observation of ADP at the SUR1 consensus site is significant. This finding, together with the prior demonstration that channel gating is not coupled to ATP hydrolysis (i.e. gating never appears to violate microscopic reversibility), supports the idea that SUR1 is an ADP sensor (**Aittoniemi et al., 2009**; **Choi et al., 2008**; **Dunne et al., 1988**; **Dunne and Petersen, 1986**; **Gribble et al., 1997**; **Kakei et al., 1986**; **Larsson et al., 1993**; **Proks et al., 2010**; **Tantama et al., 2013**). Because the affinity of the consensus site for ADP appears to be higher than for ATP, the channel can potentially sense small changes in ADP levels even in the setting of physiological ATP concentrations. In the quatrefoil form, we observe a new interface formed between SUR1 and the Kir6.2 CTD near the ADP-binding site (**Figure 6—figure supplement 3**). Perhaps relevant to this observation, mutation of a single glycine residue to arginine at this interface, found in patients with Congenital Hyperinsulinism completely destroys the stimulatory effect of MgADP in the presence of ATP(**de Wet et al., 2012**; **Stanley et al., 2004**). Further study is required to determine whether this interaction is significant to channel gating.

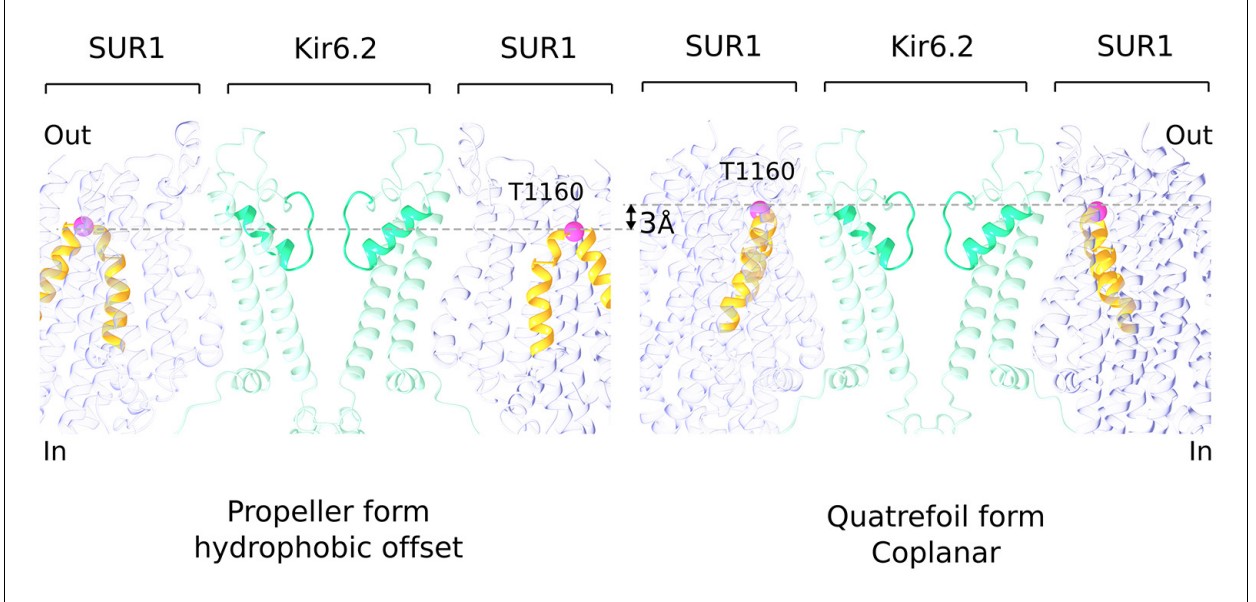

**Figure 6.** Hydrophobic offset of SUR1 relative to Kir6.2. The grey lines connect the Cα position of T1160 residues (pink sphere) located in an extracellular loop on diagonally-opposed SUR1 subunits. The pictures of propeller and quatrefoil forms are aligned to each other with respect to the Kir6.2. This representation emphasizes that the SUR1 subunits in the two structural forms are offset from each other with respect to the Kir subunit.
DOI: https://doi.org/10.7554/eLife.32481.021

The following figure supplements are available for figure 6:

**Figure supplement 1.** Comparison of the electrostatic potential surfaces of the quatrefoil and propeller forms.
DOI: https://doi.org/10.7554/eLife.32481.022

**Figure supplement 2.** Comparison of NBD-closed and NBD-open propeller forms.
DOI: https://doi.org/10.7554/eLife.32481.023

**Figure supplement 3.** The NBD2-CTD interface.
DOI: https://doi.org/10.7554/eLife.32481.024

# Materials and methods

## Key resources table

| Reagent type (species) or resource | Designation | Source or reference | Identifiers | Additional information |
|---|---|---|---|---|
| Gene (*Homo sapiens*) | SUR1 | Uniprot | Q09428 | |
| Gene (*Homo sapiens*) | Kir6.2 | Uniprot | Q14654 | |
| Cell line (Spodoptera frugiperda) | Sf9 | ATCC | | RRID: CVCL_0549 |
| Cell line (*Homo sapiens*) | HEK293S GnTI- | ATCC | | RRID: CVCL_A785 |
| Recombinant DNA reagent | pEG BacMam | doi: 10.1038/nprot.2014.173 | | |
| Software, algorithm | RELION | doi: 10.1016/j.jsb.2012.09.006 | | |
| Software, algorithm | cryoSPARC | doi: 10.1038/nmeth.4169 | | |

## Cell lines

Human embryonic kidney (HEK) cells were cultured in DMEM (GIBCO, Gaithersburg, MD) medium supplemented with 10% fetal bovine serum at (FBS) 37℃. Sf9 cells were cultured in Sf-900 II SFM medium (GIBCO) at 28℃. HEK293S GnTI⁻ cells cultured in Freestyle 293 medium (GIBCO) supplemented with 2% FBS at 37℃. Cell lines were acquired from and authenticated by American Type Culture Collection (ATCC, Manassas, VA ). The cell lines were not tested for mycoplasma contamination.

## Construct design

Synthetic cDNAs encoding human SUR1 and human Kir6.2 were initially cloned into pEG BacMam vectors containing a C-terminal GFP (*Goehring et al., 2014*). Next, we generated a fusion construct in which the C-terminus of SUR1 was linked to the N-terminus of Kir6.2 through a linker consisting 3x (Ser-Ala). To facilitate protein detection and purification, a GFP tag was appended to the C-terminus of $KATP_{em}$ with an intervening PreScission protease cleavage site and cloned into a pEG Bac-Mam vector.

## Electrophysiological recording

Human embryonic kidney (HEK) cells were cultured on coverslips placed in six-well plates. The cells in each well were transfected with synthetic cDNAs encoding human SUR1, human Kir6.2, or human $KATP_{em}$ cloned into pEG BacMam vectors containing a C-terminal GFP tag using Lipofectamine 3000 (Invitrogen, Carlsbad, CA) according to the manufacturer's instructions. After 36–48 hr, the coverslips were transferred to a recording chamber for patch clamp experiments. In whole-cell recordings, the bath solution in the recording chamber contained 5 mM Hepes, pH 7.4, 100 mM NaCl, 2.6 mM $CaCl_2$ 1.2 mM $MgCl_2$, and 40 mM KCl; micropipettes were filled with internal solution containing 5 mM Hepes, pH 7.2, 107 mM KCl, 10 mM EGTA, 1.2 mM $MgCl_2$, and 1.0 mM $CaCl_2$. For inside-out recordings, both the bath and micropipettes were filled with 10 mM Hepes, pH 7.4, 140 mM KCl, 2.6 mM $CaCl_2$ 1.2 mM $MgCl_2$. Borosilicate micropipettes were pulled and fire polished. Recording pipettes had a resistance of 2–4 MΩ. Recordings were performed at room temperature using an Axopatch 200B amplifier, a Digidata 1550 digitizer, and pCLAMP software (Molecular Devices). The recordings were low-pass filtered at 1 kHz and sampled at 20 kHz.

## Protein expression and purification

Human $KATP_{em}$ was expressed in HEK293S GnTI$^-$ cells using the BacMam method. In brief, bacmids encoding the Human $KATP_{em}$-GFP fusion were generated using DH10Bac cells according to the manufacturer's instructions. Bacmam baculoviruses were produced using *Spodoptera frugiperda* Sf9 cells cultured in SF900II SFM medium. For protein expression, suspension cultures of HEK293S GnTI$^-$ cells cultured in Freestyle 293 medium were infected with bacmam baculovirus at a density of $\sim 3 \times 10^6$ cells/ml. After 24 hr at 37°C, the infected cultures were supplemented with 10 mM sodium butyrate and grown for a further 48 hr at 30°C before harvesting. All subsequent manipulations were performed at 4°C.

For large-scale purification, cell pellets from 4 L of culture were pooled and resuspended in lysis buffer containing 50 mM Tris-Cl, pH 8.5, 500 mM KCl, 1 mM EDTA, 2 mM DTT and supplemented with a mixture of protease inhibitors (*Tao et al., 2009*). The cell suspension was subject to gentle mechanical disruption in a Dounce homogenizer and the resulting lysate was clarified at 39800 x g for 30 min. The crude membrane pellet obtained was resuspended once again in lysis buffer using a few strokes in the Dounce homogenizer and the membrane suspension was stirred for 2 hr in the presence of 1.5% (w/v) lauryl maltose neopentyl glycol (LMNG) and 0.3% (w/v) cholesteryl hemisuccinate (CHS). The solubilized membranes were clarified by centrifugation at 39800 x g for 45 min and the resulting supernatant was mixed with GFP nanobody-coupled sepharose resin (prepared in-house) by rotation. After 60 min, the resin was collected and washed with 20 column volumes of wash buffer containing 20 mM Tris-Cl, pH 8.5, 300 mM KCl, 1 mM DTT, 1 mM $MgCl_2$, 0.5 mM $CaCl_2$, 0.1% (w/v) digitonin and 0.01% (w/v) phospholipids (POPC:POPE:POPG = 3:1:1; CEG311). Elution of Human $KATP_{em}$ from the GFP nanobody resin was performed by overnight incubation with PreScission protease. The eluted protein was concentrated and then exchanged into amphipols by mixing with PMAL-C8 at amphipol:protein = 10:1 (w/w) overnight. The protein:detergent:amphipol mixture was then diluted 10-fold with detergent-free (DF) buffer (20 mM Tris-Cl, pH 8.5, 300 mM KCl) and concentrated once again to a volume of ~500 uL prior to fractionation on a Superose6 column equilibrated with DF buffer (20 mM Tris-Cl, pH 8.5, 300 mM KCl,). Peak fractions were collected and diluted to their final target concentrations before imaging by cryo-EM.

## ATPase activity assay

An NADH-coupled fluorimetric assay was used to measure ATPase activity (*Scharschmidt et al., 1979*). $Mg^{2+}$-ATP was added to a mixture containing 3.4 µM $KATP_{em}$, 50 mM Tris-Cl, pH 8.0, 300

mM KCl, 30 mM MgCl$_2$, 0.5 mM CaCl$_2$, 0.1% (w/v) digitonin and 0.01% (w/v) phospholipids (POPC: POPE:POPG = 3:1:1; CEG311), 60 µg/mL pyruvate kinase, 32 µg/mL lactate dehydrogenase, 4 mM phosphoenolpyruvate, and 300 µM NADH. Consumption of NADH was measured by monitoring the fluorescence at $\lambda_{ex}$ = 340 nm and $\lambda_{em}$ = 445 nm using an Infinite M1000 microplate reader (Tecan, Switzerland). Rates of ATP hydrolysis were calculated by subtracting the rate of fluorescence loss in the absence of KATP$_{em}$ and converting fluorescence loss to nmol NADH per minute using known standards of NADH. Data were then fit by nonlinear regression to the Michaelis-Menten equation to calculate $K_M$ and $V_{max}$ values using GraphPad Prism.

## EM data acquisition

Purified human KATP$_{em}$ in DF buffer was diluted to ~0.45 mg/mL using DF buffer and supplemented with 9 mM MgCl2, 8 mM ATP, 0.5 mM Na$_3$VO$_4$ and 150 µM C8-PIP$_2$. The supplemented protein sample was incubated at room temperature for ~3 hr and passed through a 0.45-µm filter to remove debris. To prepare cryo-EM grids, 3.5 uL drops of supplemented and filtered human KATP$_{em}$ were applied to glow-discharged Quantifoil R1.2/1.3 400 mesh Au grids. The grids were blotted for 4 s following an incubation period of 15 s at 4°C and 100% humidity before being plunge-frozen in liquid ethane using a Vitrobot Mark IV (FEI, Hillsboro, OR). The grids were imaged using a Titan Krios transmission electron microscope (FEI) operated at 300 keV. Automated data collection was performed using SerialEM (*Mastronarde, 2003*). A K2 Summit direct electron detector (Gatan, Pleasanton, CA) was used to record micrographs in super-resolution counting mode with a super-resolution pixel size of 0.65 Å. Images were recorded for 10 s over 40 frames using a dose-rate of 8 electrons per pixel per second with a defocus range of 0.9 to 2.5 µm. The total cumulative dose was ~47 electrons per Å$^2$ (~1.18 electrons per Å$^2$ per frame).

## Image processing and map calculation

Dose-fractionated super-resolution movies were 2 × 2 down-sampled by Fourier cropping to a final pixel size of 1.3 Å. The down-sampled movie frames were used for grid-based motion correction and dose-filtering with MotionCor2 (*Zheng et al., 2017*). CTF parameters were estimated from the corrected movie frames using CTFFIND4.1 (*Rohou and Grigorieff, 2015*). Motion-corrected dose-filtered movie sums were used for interactive and automated particle picking in EMAN2 (*Tang et al., 2007*). The entire dataset was also manually inspected to eliminate micrographs exhibiting imaging defects including excessive drift, cracked ice or defocus values exceeding the specified range. An initial set of 675,202 particles was obtained from 3179 images. Particle images were extracted from the motion-corrected dose-filtered images as 300 × 300 pixel boxes in RELION (*Kimanius et al., 2016*; *Scheres, 2012*).

2D classification in RELION was performed to remove spurious images of ice, carbon support and other debris, reducing the particle count to 404,205. We refer to this reduced set of particle images as the '2D filtered' particle set. *Ab initio* 3D classification was performed using stochastic gradient descent and branch-and-bound algorithms implemented in cryoSPARC (*Punjani et al., 2017*) with C4 symmetry imposed (*Figure 1—figure supplement 2*). Of the three classes obtained, two classes, containing 60% (Class A, quatrefoil form; 169,766 particles) and 18% (Class B, propeller form; 72,757) of the input particles displayed clear protein features. The remaining class was uninterpretable. The quatrefoil class was refined in cryoSPARC with a reported gold-standard FSC resolution of 3.6 Å but the quality of the density for the SUR1 transporter was rather poor. Refinement of the propeller class in cryoSPARC produced a map with reported resolution of 4.0 Å, but it too was characterized by weak density in the SUR1 transporter.

We then tested whether using a different image processing strategy could produce a better result. Using the '2D filtered' particle set as the initial input and the *ab initio* quatrefoil map from cryoSPARC (low-pass filtered to 60 Å) as the starting reference model, we performed iterative cycles of 3D refinement and 3D classification without alignment in RELION. A final round of local refinement was performed in RELION with a soft mask to exclude the amphipol belt surrounding the transmembrane portion of the particle. Over-fitting and over-estimation of resolution introduced by masking was estimated by the high-resolution noise substitution procedure implemented in RELION and the resulting corrected unbiased gold-standard FSC resolution estimates are reported. This procedure

resulted in a quatrefoil form reconstruction at FSC = 0.143 resolution of 3.9 Å (Class 1; 47282 particles) after masking (*Figure 1—figure supplements 2*, *3A*).

We observed improved helical density in the transporter module of SUR1 in Class 1 compared to the quatrefoil class from cryoSPARC (Class A). However, the density in the SUR1 NBDs remained weak and poorly resolved, which indicated structural heterogeneity and breakdown of symmetry in that region. Recent advances in statistical image processing methods allow the classification of structural heterogeneity and the recovery of improved local 3D information from inhomogeneous samples exhibiting pseudo-symmetry (*Scheres, 2016*; *Zhou et al., 2015*). To improve the Class 1 reconstruction, we divided the cryo-EM density map into eight overlapping sectors for focused 3D classification and focused 3D refinement in RELION (*Figure 1—figure supplement 4*). The sectors chosen are follows: (1) Kir6.2 CTD tetramer; (2) Kir6.2 channel tetramer; (3) Kir6.2 channel tetramer and four TMD0; (4) Kir6.2 channel tetramer, four TMD0 and one transporter module (TMD1, TMD2, NBD1 and NBD2); (5) two TMD0 and one transporter module; (6) one transporter module; (7) one CTD tetramer and one NBD dimer (NBD1 + NBD2); (8) one NBD dimer. Soft masks corresponding to these eight sectors were created using `relion_mask_create` from the RELION package. For sectors 1 to 3, C4 symmetry was enforced during focused 3D classification (without alignment) and focused 3D refinement in RELION. For sectors 4 to 8, the Class 1 particle stack were first artificially expanded 4-fold according to C4 circular symmetry using the RELION command `relion_parti-cle_symmetry_expand` (*Scheres, 2016*). The symmetry expanded particle stack was then used as input for masked 3D classification and masked 3D refinement in RELION with the focus masks corresponding to sectors 4 to 8. Masked 3D classification was performed without alignment and masked local refinement was performed to optimize alignment parameters. The masked 3D classification and refinement steps were iterated until no further improvement to the reconstructions was observed by manual inspection. For all of the focus maps obtained by this approach, we found significant enhancement to observable features and reported resolution compared to the initial map (*Figure 1—figure supplement 3*). The quality of the focus maps allowed the identification and modeling bound ligands including ATP, $Mg^{2+}$-ATP and $Mg^{2+}$-ADP as described in the main text. To facilitate map interpretation, we merged focus maps corresponding to all eight sectors into a single composite map using the compositing algorithm implemented in REFMAC5 (*Murshudov, 2016*). The resultant composite map encompasses the Kir6.2 channel tetramer, four TMD0 and one transporter module, which is equivalent to the volume enclosed in sector 4. The final composite map was sharpened by scaling to the synthetic map calculated from the refined atomic model using `diff-map.exe` (http://grigoriefflab.janelia.org/diffmap). The half-map equivalent of the composite map was prepared according to the procedure described above except half-maps from focused 3D refinements were used in lieu of the full-maps. To generate the symmetrized composite map shown in *Figure 1B* and *Figure 1—figure supplement 3B*, the region corresponding to the human KATP$_{em}$ construct (one SUR1 subunit followed by one Kir6.2 subunit) was isolated from the sharpened composite map using UCSF chimera (*Pettersen et al., 2004*) and symmetrized using `e2proc3d.py` (*Tang et al., 2007*). Local resolution was estimated using `blocres` with a box size of 18 (*Heymann and Belnap, 2007*). Representative sections of the cryo-EM density in the quatrefoil form reconstruction are shown in *Figure 1—figure supplement 5*. Finally, we compared the symmetrized composite map with the map prior to masked 3D refinement and symmetry expansion (*Figure 1—figure supplement 3A and B*). The FSC between the two maps shows a correlation at FSC = 0.5 to 6.3 Å indicating that the symmetry expansion, focused refinement and map compositing steps did not introduced global distortions (*Figure 1—figure supplement 3C*).

We applied a similar focused 3D classification and refinement with symmetry expansion to improve the initial propeller form reconstruction from cryoSPARC (Class B, see *Figure 1—figure supplement 2*). Briefly, particles corresponding to the propeller form Class B were subjected to additional rounds of *ab initio* 3D classification. A single class corresponding to 16,070 of the starting 72,757 particles displayed improved protein features in the SUR1 portion of the particle. This reconstruction was low-pass filtered to 60 Å and was used as the initial model to refine the stack of 16,070 particles in RELION, which resulted in a reconstruction at FSC = 0.143 resolution of 5.6 Å after masking (*Figure 1—figure supplement 2*). To further improve the propeller reconstruction, iterative focused 3D classification and refinement was performed in RELION using focus masks covering two overlapping sectors: (1) one SUR1 subunit (2) Kir6.2 channel tetramer and four TMD0 domains (*Figure 1—figure supplement 6*). Symmetry expansion was performed before focused

classification and refinement in Sector 1. Representative sections of the cryo-EM density in the propeller form reconstruction are shown in *Figure 1—figure supplement 7*. Composite maps (full maps and half maps) of the propeller form were prepared according to the procedure described above for the quatrefoil form reconstruction.

## Model building and coordinate refinement

Model building was initially performed in the focus maps corresponding to the quatrefoil form because of their higher quality. The cryo-EM structure of bovine MRP1 (PDB: 5UJA) was used as a reference structure to generate the starting model for building the SUR1 transporter module. Briefly, the MRP1 model (without TMD0) was first mutated to match the SUR1 sequence using CHAINSAW (*Stein, 2008*) while keeping only side chains of conserved residues. The model was then divided into two pieces, each corresponding to one half transporter (one TMD plus one NBD). The SUR1 half transporters were then docked into EM density separately by rigid-body fitting using the fitmap function in UCSF Chimera (*Pettersen et al., 2004*) followed by manual rebuilding in Coot (*Emsley et al., 2010*). To build the SUR1 TMD0 domain, the EM density in the focus maps was of sufficient quality to allow de novo manual building in Coot. The crystal structure of GIRK (PDB: 3SYA) was used a reference structure to generate the starting model for building the Kir6.2 tetramer. CHAINSAW was again used to mutate the GIRK model to match the Kir6.2 sequence and eliminating non-conserved side chains. The Kir6.2 pore module and CTD were separated and converted to tetramers by applying the appropriate symmetry operations. The pore and CTD tetramer models were then rigid-body fitted independently into EM density in UCSF Chimera to allow for rotation of the CTD relative to the pore. The fitted Kir6.2 models were then used as starting points for manual rebuilding in Coot.

The SUR1 transporter, the TMD0 domain, and the Kir6.2 tetramer channel were first built as independent initial models by consulting all available focus maps. B-factor sharpening was performed locally in Coot 'on-the-fly' to optimize observable map features for model building. Side chains were not modeled for residues with poor density. The initial models were then merged into a single consensus model containing the Kir6.2 channel tetramer, four TMD0 domains and one transporter module. Composite maps of the quatrefoil (one full map and two half maps) were then prepared as described above. Prior to automatic real space refinement, one additional round of manual rebuilding in Coot was then performed using the consensus model and the composite full map.

Automatic real space refinement of the consensus model was performed against one of the composite half-maps using `phenix.real_space_refine` (*Adams et al., 2010*) with 4-fold symmetry imposed for the Kir6.2 channel tetramer and the four TMD0 with the application of NCS constraints. Tight secondary structure and geometric restraints were used to minimize overfitting. Manual rebuilding in Coot was alternated with automated refinement in `phenix.real_space_refine`.

For cross-validation, FSC curves were calculated between refined models and the composite half-map used for refinement (FSC$_{work}$) or the composite half-map not used at any point during refinement (FSC$_{free}$) (*Figure 1—figure supplement 3D*). These curves were inspected after each round of automated real-space refinement in `phenix.real_space_refine` to monitor the effects of overfitting.

Regions that did not allow accurate establishment of amino acid register were modeled as a poly-alanines. Regions with weak or no density were not modeled and are indicated by dashed lines in *Figure 2A*. The quality of the final model was evaluated by MolProbity (*Chen et al., 2010*) and EMRinger (*Barad et al., 2015*) (*Table 1*).

Model building into the propeller form reconstruction made use of the atomic model built into the higher quality quatrefoil reconstruction. The Kir6.2 pore, the CTD, the TMD0 and transporter module from the quatrefoil form coordinates were independently docked into the propeller form reconstruction by rigid-body fitting. The agreement between the fitted models and the propeller form EM density was excellent and required little additional adjustment. The L0-loop in SUR1, which contains the lasso helix, the lasso motif and the lasso extension was ordered in the propeller form map, in contrast to the quatrefoil form. Given that the sequence of the lasso-motif in SUR1 and MRP1 is well conserved, we reasoned that the structure of this region in SUR1 should be very similar to that observed in MRP1. To build the lasso-motif in the propeller form density, we extracted the coordinates of the lasso-motif from the bMRP1 model (residues 205 to 248) and mutated it to match the SUR1 sequence using CHAINSAW while keeping sidechains of only conserved residues. This

**Table 1.**

| | KATP with Mg$^{2+}$-ATP | |
|---|---|---|
| Data acquisition | | |
| Microscope | Titan Krios | |
| Voltage (kV) | 300 | |
| Camera | Gatan K2 Summit | |
| Camera mode | Super-resolution | |
| Defocus range (μm) | 0.9 to 2.5 | |
| Pixel size (Å) | 1.3 (super-resolution = 0.65) | |
| Movies | 3179 | |
| Frames/movie | 40 | |
| Total electron dose (e-/Å$^2$) | 47 | |
| Exposure time (s) | 10 | |
| Dose rate (e$^-$/pixel/s) | 8 | |
| | Quatrefoil form | Propeller form |
| Reconstruction | | |
| Software | RELION | RELION |
| Symmetry | C4 | C4 |
| Particle number | 47,282 | 16,070 |
| Resolution (masked, Å) | 3.9 | 5.6 |
| Resolution range after focused refinement (masked, Å) | 4.1 to 3.3 | 4.5 to 3.8 |
| Model statistics | | |
| Residues built | 1596/1977 | 1667/1977 |
| Map CC (masked) | 0.825 | 0.743 |
| Cβoutliers | 0 | 0 |
| Molprobity score | 2.18 | 2.33 |
| EMRinger score | 1.82 | 0.95 |
| Ramachandran | | |
| Favored (%) | 92.48 | 90.99 |
| Allowed (%) | 7.51 | 8.73 |
| Outliers (%) | 0.00 | 0.28 |
| RMS deviations | | |
| Bond length | 0.007 | 0.006 |
| Bond angles | 1.074 | 1.134 |

DOI: https://doi.org/10.7554/eLife.32481.025

initial model (corresponding to residues 215 to 255 in SUR1) was docked into the lasso motif of the propeller form EM map by rigid-body fitting. The resulting model showed good agreement with the lasso density and required little additional manual adjustment in coot. EM density corresponding to bulky sidechains in this region allowed us to confidently establish the amino-acid register of the lasso-motif. Regions with less defined densities were built as polyalanine models. Automated real-space refinement, manual re-building and cross-validation were performed as described above for the quatrefoil form structure. The quality of the final propeller form model was evaluated by MolPro-bity and EMRinger (*Table 1*).

Structure figures were generated using UCSF ChimeraX (*Goddard et al., 2018*), PYMOL and HOLLOW (*Ho and Gruswitz, 2008*).

## Acknowledgements

We thank M Ebraham and J Sortis at the Evelyn Gruss Lipper Cryo-EM Resource Center at Rockefeller University for assistance in data collection, members of the MacKinnon and Chen laboratories for helpful discussions. KL was supported by the Human Frontiers Science Program Fellowship. RM and JC are investigators in the Howard Hughes Medical Institute.

## Additional information

### Funding

| Funder | Author |
| --- | --- |
| Human Frontier Science Program | Kenneth Pak Kin Lee |
| Howard Hughes Medical Institute | Roderick MacKinnon<br>Jue Chen |

The funders had no role in study design, data collection and interpretation, or the decision to submit the work for publication.

### Author contributions

Kenneth Pak Kin Lee, Conceptualization, Formal analysis, Investigation, Methodology, Writing—original draft, Project administration, Writing—review and editing; Jue Chen, Roderick MacKinnon, Conceptualization, Supervision, Writing—original draft, Writing—review and editing

### Author ORCIDs

Jue Chen https://orcid.org/0000-0003-2075-4283
Roderick MacKinnon http://orcid.org/0000-0001-7605-4679

### Decision letter and Author response

Decision letter https://doi.org/10.7554/eLife.32481.036
Author response https://doi.org/10.7554/eLife.32481.037

## Additional files

### Supplementary files

• Transparent reporting form
DOI: https://doi.org/10.7554/eLife.32481.026

### Major datasets

The following datasets were generated:

| Author(s) | Year | Dataset title | Dataset URL | Database, license, and accessibility information |
| --- | --- | --- | --- | --- |
| Roderick MacKinnon, Kenneth Pak Kin Lee, Jue Chen | 2018 | Cryo-EM structure of human KATP bound to ATP and ADP in quatrefoil form | http://www.rcsb.org/pdb/search/structid-Search.do?structureId=6C3O | Publicly available at the RCSB Protein Data Bank (accession no. 6C3O) |
| Roderick MacKinnon, Kenneth Pak Kin Lee, Jue Chen | 2018 | Cryo-EM structure of human KATP bound to ATP and ADP in propeller form | http://www.rcsb.org/pdb/search/structid-Search.do?structureId=6C3P | Publicly available at the RCSB Protein Data Bank (accession no. 6C3P) |
| Roderick MacKinnon, Kenneth Pak Kin Lee, Jue Chen | 2018 | Cryo-EM structure of human KATP bound to ATP and ADP in quatrefoil form | http://www.ebi.ac.uk/pdbe/entry/emdb/EMD-7338 | Publicly available at the Electron Microscopy Data Bank (accession no. EMD-7338) |

| Roderick MacKinnon, Kenneth Pak Kin Lee, Jue Chen | 2018 | Cryo-EM structure of human KATP bound to ATP and ADP in propeller form | http://www.ebi.ac.uk/pdbe/entry/emdb/EMD-7339 | Publicly available at the Electron Microscopy Data Bank (accession no. EMD-7339) |

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
