## [Decision Letter]

Thank you for submitting your article "Molecular structure of human KATP in complex with ATP and ADP" for consideration by *eLife*. Your article has been reviewed by three peer reviewers, and the evaluation has been overseen by Kenton Swartz as the Reviewing Editor and Richard Aldrich as the Senior Editor. The following individuals involved in review of your submission have agreed to reveal their identity: John L Rubinstein (Reviewer #1); Colin G. Nichols (Reviewer #2); Cristina Paulino (Reviewer #3).

The reviewers have discussed the reviews with one another and the Reviewing Editor has drafted this decision to help you prepare a revised submission.

Summary:

The manuscript by Lee et al. report a 3.9 Å and 5.6 Å cryo-EM structure of the nucleotide-bound human pancreatic KATP channel. The complex is composed of a SUR1, an ABC transporters which acts as regulator with no transport activity, and the inward-rectifying potassium channel Kir6.2. The resolution of the map was further improved by focused classification and refinement of sub-sections of the complex, which were later combined to generate a composite map. This is a rather 'recent' approach, which the authors have conducted in a thorough and excellent way, giving this study, apart from the biological impact, also high-standard methodological impact. The authors identified two conformational states from the same sample, referred to as the quatrefoil and propeller state. In both states ATP occupies the inhibitory site in Kir6.2, while the nucleotide-binding domains of SUR1 are dimerized with Mg^2+^-ATP in the degenerated site and Mg^2+^-ADP in the consensus site. The manuscript, thus, reveals a new conformation of the complex and provides new exciting details. In light of the high pharmacological relevance of the protein and its implication in diabetes, the study greatly advances our understanding of the mechanism of action of the KATP channels and will be of highest interest to several research fields. While we believe the results are of great impact and strongly support its publication in *eLife*, the manuscript requires a more comprehensive description and discussion. In particular, it will be crucial to contrast the results with the recently accepted paper in *eLife* from Martin et al., which reports the structure of the same protein in another conformation. In fact, it will give the authors the opportunity to draw a more detailed picture of the entire transport and regulation cycle, than was possible before.

Essential revisions:

1) The authors should provide a more thorough description of both maps. While the quatrefoil structure is described in detail, a similar level of detail is missing for the other propeller map. Also, a more detailed comparison of both maps would be highly appreciated, both in text form and pictures.

2) The authors propose that the Lo loop (lasso extension) interaction with the Kir6.2 cytoplasmic domain exerts strain to keep the transport module in a position where the hydrophobic transmembrane domain is shifted down with respect to the pore and TM0 domains and that the release of the interaction is triggered by MgADP-induced NBD-dimerization, which would restore hydrophobic match among the domains, resulting in reduction of ATP affinity in the Kir domain. Figure 6 is provided as supporting evidence for this hydrophobic mismatch; grey horizontal lines were drawn to designate the inferred boundaries of the membrane bilayer that supposedly corresponded the hydrophobic region of each domain. It is unclear to us whether the lines are positioned appropriately and whether such a drastic hydrophobic mismatch is present in the structure of the propeller form or not. The upper line for the pore and TM0 domain are drawn far above the more likely hydrophobic boundary and the lower line for the transporter module in the 'Propeller' form is drawn much lower than the most reasonable boundary. Even though the authors claim that the 'Quatrefoil' form has better hydrophobic matches, it seems rather opposite in fact, with the basic residues at the end of elbow1 (circled by red dots) deeply penetrating into the hydrophobic core of the bilayer. It is also worth noting that, as reported in a recent Nature paper (Norimatsu Y. et al. 2017), the bilayer surface adjacent to transmembrane proteins may be heavily perturbed by proteins. Assuming a flat bilayer surface and arguing hydrophobic mismatch may not be reasonable. Overall, the authors need to enhance their presentation and justification of the hydrophobic mismatch in order to be convincing. It would also be of interest to see how the belt of amphipols looks like in both cryo-EM maps, and if a 'micelle' distortion is visible.

3) The authors argue that the slide helix might push the neighboring PIP2 site and hence inhibit PIP2 binding. However, aligning by the whole molecule shows that the relative distances between the slide helix and the neighboring PIP2 site are more or less the same in the three structures. Pushing of the slide helix is only apparent because of rigid body rotation of the cytoplasmic domain in a counter clockwise direction if viewed from the top of the TM domain and hence the actual distances between the two parts of the Kir domain do not change since the same degree of rotation also occurs in the neighboring PIP2 site. The primary conformational change observed in these EM structures seems to be that the Kir domain is displaced from the TM domain. The displacement is greater in the quatrefoil than the propeller form. It may be more likely that loss of the canonical PIP2 binding site (lack of helical c-linker) is caused by Kir domain disengagement from the TM domain. The authors should carefully reconsider their conclusions, provide a discussion of these issues, and either modify their conclusions or enhance the presentation to more clearly communicate their conclusions.

4) Issues to discuss:a) Chicken or egg problem: The two conformations were obtained in identical conditions with the same protein and ligand combination. The authors argue that transition from the propeller to the quatrefoil conformation is triggered by Mg-ADP binding. No clear rationale is given to support this claim. Is there any additional information in the population distributions?

b) Multiple ABC transporter structures have been determined including another KATP channel (5TWV) and two-bound Mg-ADP complex (5L22). These previous structures should be considered in comparison. The authors should also compare in their new structures with the new high-resolution structure of Martin et al. in *eLife*. Of great interest is the similarity of the propeller form to the structure of Martin et al. Now that several new structures are available the authors should consider drawing a scheme for the gating mechanism, Similarly, movies provide a perfect tool to illustrate the observed conformational changes.

5) The authors state that in SUR1, ATP is found in the catalytically inactive site and ADP in the catalytically active site. We are not sure how strongly this claim can be made at the reported resolution (and from looking at 3C&D). The authors should better justify this conclusion.

---

## [Author Response]

Essential revisions:1) The authors should provide a more thorough description of both maps. While the quatrefoil structure is described in detail, a similar level of detail is missing for the other propeller map. Also, a more detailed comparison of both maps would be highly appreciated, both in text form and pictures.

In the revision, we have added focus maps and regional cryo-EM densities for the propeller form now to accompany the original quatrefoil maps (Figure 1—figure supplement 5 to 7). In addition, we also include a new figure with detailed structural comparisons of the two forms (Figure 2—figure supplement 1). In the text, we discussed the structural comparison in subsection “Two structures of human KATP”.

2) The authors propose that the Lo loop (lasso extension) interaction with the Kir6.2 cytoplasmic domain exerts strain to keep the transport module in a position where the hydrophobic transmembrane domain is shifted down with respect to the pore and TM0 domains and that the release of the interaction is triggered by MgADP-induced NBD-dimerization, which would restore hydrophobic match among the domains, resulting in reduction of ATP affinity in the Kir domain. Figure 6 is provided as supporting evidence for this hydrophobic mismatch; grey horizontal lines were drawn to designate the inferred boundaries of the membrane bilayer that supposedly corresponded the hydrophobic region of each domain. It is unclear to us whether the lines are positioned appropriately and whether such a drastic hydrophobic mismatch is present in the structure of the propeller form or not. The upper line for the pore and TM0 domain are drawn far above the more likely hydrophobic boundary and the lower line for the transporter module in the 'Propeller' form is drawn much lower than the most reasonable boundary. Even though the authors claim that the 'Quatrefoil' form has better hydrophobic matches, it seems rather opposite in fact, with the basic residues at the end of elbow1 (circled by red dots) deeply penetrating into the hydrophobic core of the bilayer. It is also worth noting that, as reported in a recent Nature paper (Norimatsu Y. et al. 2017), the bilayer surface adjacent to transmembrane proteins may be heavily perturbed by proteins. Assuming a flat bilayer surface and arguing hydrophobic mismatch may not be reasonable. Overall, the authors need to enhance their presentation and justification of the hydrophobic mismatch in order to be convincing. It would also be of interest to see how the belt of amphipols looks like in both cryo-EM maps, and if a 'micelle' distortion is visible.

We have produced a new Figure 6 (to go with Figure 6—figure supplement 1) to demonstrate the point we are making. We have also modified the associate text (subsection “Implications for channel regulation”) to explain this point clearly.

3) The authors argue that the slide helix might push the neighboring PIP2 site and hence inhibit PIP2 binding. However, aligning by the whole molecule shows that the relative distances between the slide helix and the neighboring PIP2 site are more or less the same in the three structures. Pushing of the slide helix is only apparent because of rigid body rotation of the cytoplasmic domain in a counter clockwise direction if viewed from the top of the TM domain and hence the actual distances between the two parts of the Kir domain do not change since the same degree of rotation also occurs in the neighboring PIP2 site. The primary conformational change observed in these EM structures seems to be that the Kir domain is displaced from the TM domain. The displacement is greater in the quatrefoil than the propeller form. It may be more likely that loss of the canonical PIP2 binding site (lack of helical c-linker) is caused by Kir domain disengagement from the TM domain. The authors should carefully reconsider their conclusions, provide a discussion of these issues, and either modify their conclusions or enhance the presentation to more clearly communicate their conclusions.

We are conveying the fact that the PIP2 sites are different in KATP (both forms) compared to GIRK. We have made a new figure (Figure 5—figure supplement 1), which shows clearly the difference in the PIP2 binding sites. We also modified the legend to Figure 5 to clarify the point of reference used in the structural alignment.

4) Issues to discuss:a) Chicken or egg problem: The two conformations were obtained in identical conditions with the same protein and ligand combination. The authors argue that transition from the propeller to the quatrefoil conformation is triggered by Mg-ADP binding. No clear rationale is given to support this claim. Is there any additional information in the population distributions?

We can’t find where we made this argument. We hypothesized that dimerization of the NBDs acts through the lasso extension to regulate the channel. We have no more information on this but it is an area of current study.

b) Multiple ABC transporter structures have been determined including another KATP channel (5TWV) and two-bound Mg-ADP complex (5L22). These previous structures should be considered in comparison. The authors should also compare in their new structures with the new high-resolution structure of Martin et al. in eLife. Of great interest is the similarity of the propeller form to the structure of Martin et al. Now that several new structures are available the authors should consider drawing a scheme for the gating mechanism, Similarly, movies provide a perfect tool to illustrate the observed conformational changes.

We added a comparison of our propeller form to one previously published (Figure 6—figure supplement 2), as requested. We have also compared and discussed the structure of 5L22 (structure with Mg-ADP) (subsection “The nucleotide bound state of SUR1” and Figure 3—figure supplement 2). We think it is premature to draw out a gating mechanism.

5) The authors state that in SUR1, ATP is found in the catalytically inactive site and ADP in the catalytically active site. We are not sure how strongly this claim can be made at the reported resolution (and from looking at 3C&D). The authors should better justify this conclusion.

In the revision we have now included a new figure (Figure 3—figure supplement 1) comparing the nucleotide densities in the consensus and degenerate nucleotide binding sites. These panels clearly demonstrate that EM density corresponding to the γ-phosphate of ATP is present in the degenerate site but is absent in the consensus site. As you will see, the density is of high quality in this region.